# Targeting IKKβ Activity to Limit Sterile Inflammation in Acetaminophen-Induced Hepatotoxicity in Mice

**DOI:** 10.3390/pharmaceutics15020710

**Published:** 2023-02-20

**Authors:** Song-Hee Kim, Da-Eun Jung, Jin Yong Song, Jihye Jung, Jae-Kyung Jung, Heesoon Lee, Eunmiri Roh, Jin Tae Hong, Sang-Bae Han, Youngsoo Kim

**Affiliations:** 1College of Pharmacy, Chungbuk National University, Cheongju 28160, Republic of Korea; 2Department of Cosmetic Science, Kwangju Women’s University, Gwangju 62396, Republic of Korea

**Keywords:** acetaminophen, hepatotoxicity, damage-associated molecular pattern, sterile inflammation, IKKβ inhibitor

## Abstract

The kinase activity of inhibitory κB kinase β (IKKβ) acts as a signal transducer in the activating pathway of nuclear factor-κB (NF-κB), a master regulator of inflammation and cell death in the development of numerous hepatocellular injuries. However, the importance of IKKβ activity on acetaminophen (APAP)-induced hepatotoxicity remains to be defined. Here, a derivative of caffeic acid benzylamide (CABA) inhibited the kinase activity of IKKβ, as did IMD-0354 and sulfasalazine which show therapeutic efficacy against inflammatory diseases through a common mechanism: inhibiting IKKβ activity. To understand the importance of IKKβ activity in sterile inflammation during hepatotoxicity, C57BL/6 mice were treated with CABA, IMD-0354, or sulfasalazine after APAP overdose. These small-molecule inhibitors of IKKβ activity protected the APAP-challenged mice from necrotic injury around the centrilobular zone in the liver, and rescued the mice from hepatic damage-associated lethality. From a molecular perspective, IKKβ inhibitors directly interrupted sterile inflammation in the Kupffer cells of APAP-challenged mice, such as damage-associated molecular pattern (DAMP)-induced activation of NF-κB activity via IKKβ, and NF-κB-regulated expression of cytokines and chemokines. However, CABA did not affect the upstream pathogenic events, including oxidative stress with glutathione depletion in hepatocytes after APAP overdose. *N*-acetyl cysteine (NAC), the only FDA-approved antidote against APAP overdose, replenishes cellular levels of glutathione, but its limited efficacy is concerning in late-presenting patients who have already undergone oxidative stress in the liver. Taken together, we propose a novel hypothesis that chemical inhibition of IKKβ activity in sterile inflammation could mitigate APAP-induced hepatotoxicity in mice, and have the potential to complement NAC treatment in APAP overdoses.

## 1. Introduction

Acetaminophen (APAP) is a safe and effective analgesic and fever medication in humans. At therapeutic doses, APAP is subjected to glucuronidation or sulfation in the liver, and is excreted in urine [1]. However, APAP overdose causes hepatotoxicity and hepatic failure [2]. Excessive APAP is metabolized to *N*-acetyl-*p*-benzoquinone imine (NAPQI) by cytochrome P450 enzymes, such as CYP2E1 and CYP1A2, in hepatocytes [1,2]. NAPQI is detoxified by conjugation with glutathione, and this process involves glutathione-*S*-transferases such as GSTP1 and GSTT1 [1,2]. High concentrations of NAPQI deplete glutathione stores in the liver when the GST enzymes are overwhelmed [1,2]. Excessive NAPQI reacts with the thiol groups in cellular proteins, especially in mitochondrial proteins, leading to the formation of covalent adducts that initiate APAP-induced hepatotoxicity [1,2].

A major consequence of NAPQI-induced protein adducts after APAP overdose causes oxidative and nitrosative stress in the mitochondria of hepatocytes [3,4]. This compromises the leakage of electrons from the electron transport chain in the mitochondria, generating reactive oxygen species (ROS) such as superoxide [3,4]. Endogenous nitric oxide reacts with superoxide radicals to produce highly reactive peroxynitrite, resulting in nitrosative stress [3,4]. APAP-induced hepatotoxicity downstream of mitochondrial stress is further exaggerated by sustained activation of c-Jun-*N*-terminal kinase (JNK) in the cytosol [3,4]. Redox-sensitive protein kinases such as apoptosis signaling regulating kinase 1 (ASK1) and mixed-linkage kinase 3 (MLK3) are self-activated through dissociation with thioredoxin, which phosphorylates mitogen-activated protein kinase kinase 4/7 (MKK4/7) [3,4]. MKK4/7, in turn, catalyzes the phosphorylation of JNK at Ser-176 and Ser-180 residues, which is an activation index of JNK activity [3,4]. Phospho (p)-JNK is translocated to the mitochondria, where it binds to and phosphorylates SH3 homology-associated BTK-binding protein (Sab) [3,4]. Furthermore, p-JNK phosphorylates 14-3-3, a protein that sequesters Bax in the cytosol, leading to the translocation and insertion of Bax into the mitochondrial outer membrane [3,4]. Both p-JNK and Bax, once translocated to the mitochondria, trigger the opening of mitochondrial permeability transition pores, collapse of membrane potential, and ATP depletion [3,4,5]. These events facilitate the release of mitochondrial intermembrane proteins, endonuclease G, and apoptosis-inducing factor (AIF), which are translocated to the nucleus, inducing the genomic DNA fragmentation in hepatocyte necrosis [3,4,6].

Sterile inflammation is a host defense mechanism in the absence of over infection, and its priming signal is likely to occur when damage-associated molecular patterns (DAMPs) are passively released from dying or damaged hepatocytes after APAP overdose [7]. DAMPs are endogenous substances such as high mobility group box 1 (HMGB1), heat shock proteins (HSPs), and DNA fragments [7,8]. After escaping from their normal cellular compartments in hepatocytes, DAMPs bind to Toll-like receptors (TLRs) in Kupffer cells, the resident liver macrophages [7,8]. This event invokes a pro-inflammatory phenotype in Kupffer cells, stimulating the expression of cytokines and chemokines, and the production of other inflammatory mediators [7,9]. Therefore, targeting a single DAMP molecule may only have a limited effect on the blockade of sterile inflammation during APAP-induced hepatotoxicity. Kupffer cell-secreted cytokines and chemokines recruit leukocytes, including neutrophils and monocyte-derived macrophages, into the liver, which sustains the progression of the inflammatory responses [7,10]. The infiltrated immune cells mainly contribute to the removal of necrotic cell debris in the liver and promote tissue repair in preparation for hepatocyte regeneration [7,11]. In contrast, if not appropriately controlled, the phagocytic cells can provoke or exacerbate tissue injury in APAP-induced hepatotoxicity [7].

*N*-Acetyl cysteine (NAC) is the only FDA-approved antidote against APAP overdose [12]. As a mechanism, NAC replenishes cellular glutathione levels, thus enhancing the detoxification of NAPQI in the cytoplasm and scavenging ROS in the mitochondria to support energy metabolism [12]. Although NAC is highly effective when given at early time-points after APAP overdose, its limited efficacy is concerning in late-presenting patients who have already undergone oxidative stress in the liver in the earlier stage of pathogenesis [12].

The kinase activity of inhibitory κB kinase β (IKKβ) acts as a signal transducer in the activating pathway of nuclear factor-κB (NF-κB), a master regulator of inflammation and cell death in the development of numerous hepatocellular injuries [13]. The importance of IKKβ activity on APAP-induced hepatotoxicity remains to be defined, even though mice with hepatocyte-specific knockout of IKKα, encoding another catalytic activity of IKK complex, repress APAP-induced hepatic injury and mortality [14]. In this study, a derivative of caffeic acid benzylamide (CABA, Figure 1A) was found to inhibit the kinase activity of IKKβ, as did IMD-0354 and sulfasalazine. IMD-0354 is a selective inhibitor of IKKβ with shows anti-inflammatory activity by lowing NF-κB activation and is a candidate drug in clinical trials for atopic dermatitis [15]. Sulfasalazine is an FDA-approved anti-inflammatory drug used to treat rheumatoid arthritis and ulcerative colitis [16]. C57BL/6 mice were treated with CABA, IMD-0354, or sulfasalazine after APAP overdose. These small-molecule inhibitors of IKKβ activity protected the APAP-challenged mice from necrotic injury in the liver, and rescued them from mortality. From a molecular perspective, the IKKβ inhibitors directly mitigated sterile inflammation through suppression of DAMP-induced activation of NF-κB activity and subsequent NF-κB-regulated expression of cytokines and chemokines in Kupffer cells.

## 2. Materials and Methods

### 2.1. Chemical Synthesis of CABA

CABA (>97% purity) was synthesized by amidation reaction of caffeic acid. To a solution of caffeic acid (1.0 equiv.), HOBt (1.0 equiv.) and EDCI (1.0 equiv.) in DMF were added and stirred at room temperature for 15 min. To this solution, amine derivative (1.0 equiv.) was added and the reaction mixture was stirred at room temperature. After 15 min, the reaction mixture was treated with DIPEA (2.0 equiv.) and stirred at room temperature for 16 h. The reaction progress was monitored by TLC and then it was quenched with 1N HCl and extracted with EtOAc (3 × 15 mL); the organic layer was dried over MgSO_4_, filtered, and concentrated in vacuo. The residue was purified by silica gel column chromatography (10–25% EtOAc/hexanes) to obtain CABA in good yields (76%). IR 3299, 2926, 1649, 1589, 1442, 1112 cm^−1^; ^1^H NMR (CD_3_OD, 400 MHz) δ 7.74, 7.61 (d, 5H, J = 7.2, 7.2 Hz), 7.20 (s, 1H), 7.06 (d, 1H, J = 7.6 Hz), 6.93 (d, 1H, J = 7.6 Hz), 6.61 (d, 1H, J = 14.8 Hz), 4.69 (s, 2H); ^13^C NMR (CD_3_OD, 100 MHz) δ 168.0, 147.5, 145.3, 143.4, 141.5, 129.1, 126.8, 125.7, 123.0, 120.8, 116.6, 115.1, 113.7, 42.3; LC-MS *m*/*z* calculated for C_17_H_14_F_3_NO_3_ [M-H]^−^ = 336.29; found 336.1.

### 2.2. Antibodies and Pharmacological Agents

Anti-p-NF-κB p65, anti-NF-κB p65, anti-p-c-Jun, anti-p-interferon regulatory factor 3 (p-IRF3), anti-IRF3, anti-p-IKKα/β, anti-GAPDH, anti-nuclear erythroid-related factor 2 (NRF2), anti-p53 tumor suppressor (p53), anti-p-JNK, and anti-JNK were purchased from Cell Signaling, Danvers, MA, USA; anti-c-Jun and anti-IKKα/β from Santa Cruz, Dallas, TX, USA; and anti-HSP70 from Abcam, Cambridge, UK. Horseradish peroxidase (HRP)-conjugated rabbit anti-goat IgG, HRP-conjugated goat anti-rabbit IgG, and HRP-conjugated goat anti-mouse IgG were purchased from ThermoFisher, Waltham, MA, USA. APAP, sulfasalazine, and NAC were purchased from Sigma-Aldrich, St. Louis, MO, USA; IMD-0354 from TOCRIS, Bristol, UK; rhHMGB1 from R&D Systems, Minneapolis, MN, USA; and rhHSP70 from ENZO, New York, NY, USA.

### 2.3. Cell-Free Kinase Assay of IKKβ or TAK1

rhIKKβ was purchased from Merck Millipore, Burlington, MA, USA; and rhTAK1-TAB1 from Promega, Madison, WI, USA. These catalytically active enzymes were reacted with major basic protein (MBP) as the exogenous substrate in the presence of 5 μCi [γ-^32^P]-labeled ATP (Perkin Elmer) at 30 °C for 30 min. The reaction mixtures were spotted onto a P81 phosphocellulose filter, washed with 0.75% H_3_PO_4_, and washed again with 100% acetone. Kinase activity was measured as count per minute (cpm) of radioactivity.

### 2.4. Hepatic Injury Model in Mice

C57BL/6 mice (male, 7 weeks old) were purchased from OrientBio, Seongnam, Republic of Korea and housed at controlled conditions of temperature (22 ± 2 °C), humidity (55 ± 5%), and a 12 h/12 h light–dark cycle. The mice were fasted for approximately 12 h with free access to water and then administered APAP (500 mg/kg, dissolved in warm phosphate-buffered saline) via oral gavage to induce liver injury. Optimal doses of IKKβ inhibitors were determined through preliminary experiments. CABA (30 mg/kg or 100 mg/kg), IMD-0654 (30 mg/kg), sulfasalazine (50 mg/kg), or NAC (500 mg/kg) was dissolved in phosphate-buffered saline supplemented with 0.5% ethyl alcohol and 0.5% Tween 80, and intravenously injected into the mice at 1 h after APAP overdose according to the experimental protocol (Figure 1B). Blood was collected by direct cardiac puncture. The serum was analyzed for the levels of aminotransferases (AST, ALT) and bilirubin. Liver lobules were excised from the mice. For histological analysis, liver tissues around the centrilobular zone were fixed in 4% paraformaldehyde, embedded in paraffin, and sliced into sections with a thickness of 5 μm. These liver specimens were de-paraffinized and stained with hematoxylin and eosin (H&E; ThermoFisher, Sigma-Aldrich) or subjected to the TUNEL assay (Roche, Basel, Switzerland).

### 2.5. Co-Culture Model between Primary Hepatocytes and Kupffer Cells

Primary hepatocytes and Kupffer cells were isolated from healthy livers of C57BL/6 mice as described previously [17,18]. As shown in Figure 1C, primary hepatocytes (4 × 10^5^) were seeded on collagen-coated plates and pre-incubated for 6 h. Kupffer cells (7 × 10^5^) were seeded on trans-well inserts with a pore size of 0.4 μm (Corning, New York, NY, USA) and pre-incubated for 12 h. Trans-well inserts containing Kupffer cells were transferred onto the plate of hepatocytes, and cultured in RPMI 1640 medium (Gibco, Billings, MT, USA) supplemented with 10% fetal bovine serum (Corning), according to the experimental protocol (Figure 1C). ImKC, an immortalized mouse Kupffer cell line (Merck-Millipore), was cultured in RPMI 1640 medium supplemented with 10% fetal bovine serum in an atmosphere of 5% CO_2_ at 37 °C.

### 2.6. Western Blot Analysis

Protein extracts were resolved using SDS-PAGE and transferred to PVDF membranes, which were then blocked with 5% skim milk before overnight incubation with primary antibodies at 4 °C. After washing with Tris-buffered saline containing Tween-20, the membranes were incubated with HRP-conjugated secondary antibodies for 1 h at room temperature and visualized using an enhanced chemiluminescence (ECL) detection kit (GE Healthcare, Chicago, IL, USA).

### 2.7. Luciferase Reporter Assay

The transcriptional activity of NF-κB was determined in ImKC cells after transfection with the NF-κB-Luc reporter in combination with a *Renilla* control vector using a Lipofectamine kit (ThermoFisher). Protein extracts were subjected to a dual-luciferase assay (Promega). Firefly luciferase activity, which reports the transcriptional activity of NF-κB, was normalized to *Renilla* activity as a reference of the transfection efficiency.

### 2.8. RT-PCR Analysis

Total RNAs were isolated using a premix kit (NucleoZoL), and cDNA was synthesized from 1 μg RNA using an RT premix kit (Intron, Seongnam, Republic of Korea). The primers used in this study are listed in Appendix A. Target cDNA was amplified with a PCR premix (Bioneer, Daejeon, Republic of Korea) under the following conditions; 25–30 cycles at 94 °C for 30 s, at 50–60 °C for 30 s, and at 72 °C for 30–60 s. The reaction product was resolved on an agarose gel by electrophoresis, and visualized using EcoDye (Biofact, Daejeon, Republic of Korea).

### 2.9. ELISA

Blood samples were centrifuged at 4 °C for 10 min at 3500× *g*. Plasma was collected and stored at −80 ℃. The levels of HSP70 and HMGB1 in the plasma were measured using ELISA kits (Enzo, IBL International, Hamburg, Germany). 

### 2.10. Data Analysis

The results are expressed as mean ± SD values (*n* = 3, unless otherwise specified). Statistical significance was determined using ANOVA and Student’s *t*-test. The values of *p* < 0.05 were considered statistically significant.

## 3. Results

### 3.1. Rescue of APAP-Induced Mortality in Mice by Small-Molecule Inhibitors of IKKβ Activity

CABA, IMD-0354, and sulfasalazine inhibited IKKβ activity in cell-free reactions (Figure 2A). IMD-0354 and sulfasalazine exhibit therapeutic efficacy against inflammatory diseases through a common mechanism by acting as competitive inhibitors of IKKβ activity with respect to the ATP-binding site [16,19].

To examine the importance of IKKβ activity in APAP-induced hepatotoxicity, C57BL/6 mice were treated with CABA, IMD-0354, or sulfasalazine at 1 h after APAP overdose, as shown in Figure 1B. The APAP-challenged mice were sacrificed in a time-dependent manner; the onset of mortality occurred at approximately 18 h after APAP administration (Figure 2B). Treatment with IKKβ inhibitors decreased the mortality of APAP-challenged mice, such that 87% of mice in the CABA (100 mg/kg)-treated group, 74% of mice in the IMD-0354 (30 mg/kg)-treated group, 55% of mice in the sulfasalazine (50 mg/kg)-treated group, and 50% of mice in the NAC (500 mg/kg)-treated group survived, whereas almost all the mice in the vehicle-treated group died within the observed period of 72 h (Figure 2B).

### 3.2. Protection against Necrotic Injury in the Liver by IKKβ Inhibitors

To assess if APAP-induced lethality was associated with hepatic damage, we examined the plasma levels of aminotransferases (ALT, AST) and total bilirubin. After APAP overdose, the CABA-, IMD-0354-, and sulfasalazine-treated mice presented significantly lower levels of ALT, AST, and bilirubin in the blood compared to the vehicle-treated group (Figure 3A). Lytic injury in parenchymal hepatocytes releases aminotransferases into the circulation, and impaired glucuronide conjugation in the liver disturbs bilirubin excretion [20].

The H&E staining results showed that the APAP-challenged mice had impaired livers, indicated by a confluent area of necrosis around the centrilobular zone and loss of cellular integrity in the plasma and organelle membranes (Figure 3B). However, the architecture of liver tissue after treatment with CABA or NAC was intact or similar to normal state, implying restoration of liver tissues (Figure 3B).

TUNEL assay results revealed that the APAP-challenged mice underwent extensive DNA fragmentation in hepatocytes (Figure 3B); DNA diffusion into the cytoplasm and extracellular spaces was visible in the areas with necrotic injury. Treatment with CABA reduced the APAP-induced increase in the numbers of TUNEL-positive cells in the liver, as did NAC (Figure 3B). APAP-induced cell death accompanies the fragmentation of genomic DNA in the absence of caspase activation and lack of protection by caspase inhibitors [6].

Moreover, increased numbers of Ly6G-postive cells were detected in the APAP-challenged mice at the necrotic sites in the liver (Appendix A), indicating neutrophil infiltration and intrahepatic migration. Treatment with CABA inhibited APAP-induced neutrophil accumulation in the necrotic lesions (Appendix A). APAP-induced hepatotoxicity represents two waves of neutrophil movements. One is the infiltration into the periphery of the injury or sinusoids in the early phase, causing additional damage and the other is the intrahepatic migration toward necrotic lesions in the delayed kinetics, which removes the injured or dead cells and facilitates the repair of the injured tissues [21].

### 3.3. Inhibition of IKKβ-Catalyzed Phosphorylation of IκBα in Kupffer Cells but Not in Hepatocytes by CABA

APAP overdose in the mice stimulated the phosphorylation of IκBα in the liver as well as that of IKKα/β (Figure 4A,B). Treatment with CABA after APAP overdose decreased the levels of p-IκBα in the liver and inhibited the subsequent degradation of the IκBα protein; NAC exerted a similar effect (Figure 4A). Whereas CABA had no effect on the levels of p-IKKα/β, NAC was effective in decreasing these levels (Figure 4B). These results indicate that the mechanism underlying CABA activity could be different from that of NAC.

To understand which parenchymal or non-parenchymal liver cells could be sensitized to IκBα phosphorylation in response to APAP overdose, we introduced a co-culture model of primary hepatocytes in combination with primary Kupffer cells under conditions that prevented direct contact between bilateral cells but allowed crosstalk via soluble factors in the media, as shown in Figure 1C. In the co-culture model, Kupffer cells enhanced the phosphorylation of IκBα upon exposure to APAP, whereas hepatocytes exhibited similar levels of p-IκBα as the unstimulated state (Figure 4C). Treatment with CABA inhibited APAP-induced phosphorylation of IκBα and the subsequent degradation of IκBα protein in Kupffer cells, but it had no effect on the levels of p-IκBα and total IκBα in hepatocytes (Figure 4C). Interestingly, monoculture of Kupffer cells did not alter the levels of p-IκBα and total IκBα even after stimulation with APAP or treatment with CABA (Figure 4D). APAP also stimulated the phosphorylation of IKKα/β in Kupffer cells but not in hepatocytes in the co-culture model (Figure 4E). Treatment with CABA did not inhibit either APAP-induced phosphorylation of IKKα/β in Kupffer cells (Figure 4E) or the kinase activity of rhTAK1-TAB1 in cell-free reactions (Appendix A), indicating that CABA could target the catalytic activity of IKKβ but not that of TAK1. Moreover, CABA at concentrations of 3–30 μM did not affect the viability of hepatocytes or Kupffer cells (Appendix A), excluding possible cytotoxicity. These results suggest that CABA could inhibit IKKβ-catalyzed phosphorylation of IκBα in Kupffer cells but not in hepatocytes of APAP-challenged mice. Here, we hypothesized that IKKβ activity in Kupffer cells might be triggered by crosstalk with soluble factors, such as DAMPs, that were secreted from hepatocytes after APAP overdose.

IKKβ catalyzes the phosphorylation of IκBα at Ser-32 and -36 residues and TAK1 phosphorylates IKKα/β at Ser-177 and -181 residues, an activation index of IKKα/β activity [13]. IKKβ is located downstream of TAK1 as a signal transducer in the canonical pathway for the activation of NF-κB activity [13]. NF-κB is normally sequestered in the cytoplasm in an inactive form through entanglement with inhibitory proteins of the IκB family [13]. IκB proteins are phosphorylated as a result of IKKβ activity following exposure to internal and external stimuli [13]. The p-IκBs are subjected to ubiquitination by Skp1-Culin-F-box-type E3 ubiquitin ligases followed by proteolytic degradation by the 26S proteasome in the cytoplasm [13,22]. Thereafter, free NF-κB translocates into the nucleus and exerts it effects on the transcriptional activity on target genes [13].

### 3.4. Interruption of DAMP-Induced Activation of NF-κB Activity in Kupffer Cells by IKKβ Inhibitors

The APAP-challenged mice triggered the phosphorylation of NF-κB p65 in the liver, which was inhibited by treatment with IKKβ inhibitors (CABA, IMD-0354, and sulfasalazine) or NAC (Figure 5A,B). In the co-culture model, Kupffer cells enhanced the phosphorylation of NF-κB p65 upon exposure to APAP, whereas hepatocytes exhibited similar levels of p-NF-κB p65 as that of the unstimulated state (Figure 5C). Treatment with CABA inhibited the APAP-induced phosphorylation of NF-κB p65 in Kupffer cells without affecting the levels of p-NF-κB p65 in hepatocytes (Figure 5C). Interestingly, Kupffer cells in the monoculture did not show altered levels of p-NF-κB p65 even after stimulation with APAP or treatment with CABA (Figure 5D). In contrast, monoculture of Kupffer cells increased the levels of p-NF-κB p65 upon exposure to conditioned media from APAP-activated hepatocytes, which was inhibited by treatment with CABA (Figure 5E). Moreover, treatment with IKKβ inhibitors such as CABA, IMD-0354, and sulfasalazine attenuated the conditioned media-induced transcriptional activity of NF-κB in ImKC cells harboring the NF-κB-Luc reporter (Figure 5F). rhHMGB1 and rhHSP70 were pre-incubated with polymyxin B for the removal of endotoxin lipopolysaccharide, and employed as surrogates of DAMPs. rhHMGB1 and rhHSP70 triggered the transcriptional activity of NF-κB in ImKC cells, which was consistently inhibited by treatment with IKKβ inhibitors (Figure 5G,H).

Moreover, the APAP-challenged mice activated the pathways of activator protein 1 (AP-1), consisting of c-Fos/c-Jun dimers, and IRF3 in the liver (Appendix A). Treatment with CABA had no effect on the APAP-induced increase in c-Fos, p-c-Jun, or p-IRF3 levels (Appendix A). These results indicate that hepatocyte-secreted DAMPs, such as HMGB1 and HSP70, could stimulate the activating pathway of NF-κB in Kupffer cells and that of AP-1 and IRF3 in the liver of APAP-challenged mice. Treatment with IKKβ inhibitors interrupted the DAMP-induced activation loop of NF-κB activity but not those of AP-1 and IRF3.

IKKβ catalyzes the phosphorylation of NF-κB p65 at the Ser-536 residue, which is an activation index of NF-κB activity [13]. Nuclear p-NF-κB p65 recruits the co-activators CBP/p300, which are involved in chromatin remodeling, and the transcription machinery containing RNA polymerase II onto the promoters of NF-κB-target genes [23]. Injured or dying hepatocytes after APAP overdose lead to the leakage of DAMPs including HMGB1 and HSP70 from their normal cellular compartments [8,24]. Thereafter, either HMGB1 or HSP70 binds to TLRs in Kupffer cells, and transmits signal cascades in the activating pathways of NF-κB, AP-1, and IRF3, resulting in the initiation of sterile inflammation [24]. IKKβ essentially functions as a signal transducer in TLR-mediated canonical activation of NF-κB activity, but not in that of AP-1 or IRF3 [24]. The transcriptional activity of AP-1 is enhanced by up-regulating c-Fos expression and phosphorylating c-Jun at Ser-63 and -73 residues [24,25]. IRF3 is phosphorylated at the Ser-396 residue in the cytoplasm, following which p-IRF3 undergoes dimerization to form IRF3/IRF3 or IRF3/IRF7 dimers, and is translocated into the nucleus [24,26].

### 3.5. Suppression of Cytokine and Chemokine Expression during Sterile Inflammation by CABA

We asked whether CABA could alter NF-κB-regulated expression of cytokines and chemokines during the progression of sterile inflammation, since IKKβ inhibitors had interrupted upstream signaling such as DAMP-induced activation of NF-κB activity, as shown in Figure 5E–H. The APAP-challenged mice up-regulated the transcription of pro-inflammatory cytokines (TNF-α, IL-1β, IL-6) as well as those of chemokines (CCL2, CXCL1, CXCL2) in the liver, which was suppressed by treatment with CABA or NAC (Figure 6A).

In the co-culture model, APAP consistently increased the mRNA levels of TNF-α and CCL2 in Kupffer cells, but not in hepatocytes (Figure 6B). Treatment with CABA down-regulated the APAP-induced transcription of TNF-α and CCL2 in Kupffer cells (Figure 6B). As expected, monoculture of Kupffer cells did not show alterations in the mRNA levels of TNF-α or CCL2 following stimulation with APAP or treatment with CABA (Figure 6C). These results suggest that CABA could suppress the transcription of TNF-α, IL-1β, IL-6, CCL2, CXCL1, and CXCL2 during sterile inflammation in the liver of APAP-challenged mice.

IKKβ activity is essentially involved in the activation of NF-κB, a transcription factor that plays pleiotropic roles in regulating the expression of many inflammatory genes [13]. NF-κB-responsive κB motifs are located at the sites −653/−643, −624/−614, and −504/−494 in the TNF-α gene; −297/−288 in the IL-1β gene; −125/−111 in the IL-6 gene; −2378/−2370 and −2352/−2344 in the CCL2 gene; −597/−587, −88/−78, and −69/−59 in the CXCL1 gene; and −97/−87 in the CXCL2 gene [27,28,29,30]. Kupffer cell-derived TNF-α and IL-1β can activate endothelial cells in sinusoids to up-regulate the expression of adhesion molecules and facilitate the hepatic entry of peripheral immune cells [10,31]. IL-6 exaggerates liver injury after APAP overdose under conditions such as low expression of anti-inflammatory cytokines [10,31]. In contrast, IL-6 stimulates hepatocyte proliferation in the repair of APAP-injured tissue by binding to its receptors gp80 and gp130, resulting in the activation of another transcription factor, STAT3, via Janus kinases [10,31]. CCL2 is a chemokine involved in the intrahepatic migration of monocyte-derived macrophages toward APAP-elicited necrotic lesions [10,31]. CXCL1 and CXCL2 recruit neutrophils to the periphery of APAP-injured tissues in the liver [10,31].

### 3.6. Inert Effectiveness on DAMP Release and Oxidative Stress in Hepatocytes by CABA

We asked whether CABA could directly affect the release of DAMPs, such as HMGB1 and HSP70, in the priming of sterile inflammation, since IKKβ inhibitors can interrupt downstream signaling and DAMP-induced activation of NF-κB activity, as shown in Figure 5E–H. The APAP-challenged mice had increased plasma levels of HMGB1 and HSP70, in blood samples collected at 15 h after APAP overdose (Figure 7A). Treatment with CABA had no effect on the APAP-induced increase in the plasma levels of HMGB1 and HSP70 (Figure 7A).

To further examine the release of DAMPs, primary hepatocytes were exposed to APAP, and their protein extracts were prepared for immunoblot analysis. The hepatocytes released HMGB1 at 3 h and HSP70 at 9 h after exposure to APAP, which was not affected by treatment with CABA (Figure 7B,C). These results indicate that CABA had no effect on the leakage of DAMPs from the hepatocytes of APAP-challenged mice.

Next, we focused on the enzymes that metabolize NAPQI in hepatocytes. Metabolism of NAPQI is critical in the initial pathogenesis of APAP overdose [1]. The APAP-challenged mice up-regulated the transcription of CYP2E1 and CYP1A2 in the liver but down-regulated that of GSTP1 and GSTT1, and treatment of CABA did not alter these responses (Figure 8A). The APAP-challenged mice had depleted hepatic levels of glutathione (Figure 8B) and sustained phosphorylation (activation) of JNK in hepatocytes (Figure 8C,D). Treatment with CABA had no effect on either APAP-induced glutathione depletion or JNK phosphorylation in the liver (Figure 8B–D). Moreover, the APAP-challenged mice had elevated levels of redox-sensitive transcription factors, such as Nrf2 and p53 in the liver, which were not affected by treatment with CABA (Appendix A). These results suggest that CABA may be ineffective in protecting APAP-challenged mice from oxidative stress in hepatocytes.

Many compensatory pro-survival signaling pathways can resolve oxidative stress in hepatocytes following APAP overdose. Normally, Nrf2 is entangled with Kelch-like ECH-associated protein 1 (Keap1) and ubiquitinated by Keap1 in juxtaposition, causing Nrf2 degradation by the 26S proteasome in the liver [32]. However, the thiols on Cys residues of Keap1 are converted to disulfide bridges under oxidative stress, undergoing a conformational change in Keap1 that weakens its binding to Nrf2, and resulting in stabilization of the Nrf2 protein [32]. The transcription factor Nrf2 up-regulates the expression of Glu-Cys ligase, which maintains hepatic levels of glutathione, and that of GSTs that accelerate the detoxification of NAPQI in hepatocytes [2]. Similarly, low levels of p53 are maintained in the liver due to ubiquitination by the mouse double minute 2 homolog (MDM2), leading to p53 degradation [33]. A conformational change in p53 through specific phosphorylation by redox-sensitive kinases dismantles the p53–MDM2 axis [33]. The transcription factor p53 regulates the expression of several genes. p53 plays a protective role against oxidative stress-induced injury of hepatocytes after APAP overdose, but it delays the restorative events such as the removal of injured or dead cells and progression of tissue repair [33,34].

## 4. Discussion

In the current study, CABA inhibited the kinase activity of IKKβ, as did IMD-0354 and sulfasalazine. Treatment with CABA, IMD-0354, or sulfasalazine protected the APAP-challenged mice from histopathologic lesions around the centrilobular zone in the liver, and rescued the mice from hepatic damage-associated lethality. The protective activity of IKKβ inhibitors against APAP-induced hepatotoxicity was evident through the mitigation of necrotic cell death in the liver and amelioration of circulating ALT, AST, and bilirubin levels. As proposed in Figure 9, unique features of IKKβ inhibitors such as CABA, IMD-0354, and sulfasalazine directly interrupted sterile inflammation in Kupffer cells but did not affect oxidative stress in hepatocytes in protecting the APAP-challenged mice from hepatic injury and mortality. Sterile inflammation is likely to occur when DAMPs are passively released from dying or damaged hepatocytes after APAP overdose followed by the binding of DAMPs to TLRs in Kupffer cells at the late stage of APAP-induced hepatotoxicity. NAC is the only FDA-approved antidote against APAP overdose but its limited efficacy is concerning in late-presenting patients who have already undergone oxidative stress in the liver during the earlier stage of pathogenesis [12]. Taken together, we propose a novel hypothesis that chemical inhibition of IKKβ activity in sterile inflammation could mitigate APAP-induced hepatotoxicity in mice, and might have a potential to complement NAC treatment as an antidote against APAP overdose.

In the co-culture model of parenchymal and non-parenchymal liver cells, primary Kupffer cells stimulated IKKβ-catalyzed phosphorylation of IκBα and NF-κB p65 in response to APAP overdose, whereas primary hepatocytes exhibited similar levels of p-IκBα and p-NF-κB p65 as in the unstimulated state. Treatment with IKKβ inhibitors attenuated APAP-induced phosphorylation of IκBα and NF-κB p65 in Kupffer cell but not in hepatocytes. In the monoculture of Kupffer cells, stimulation with APAP did not alter p-IκBα and p-NF-κB p65 levels when compared with that in the unstimulated state. In contrast, monoculture of Kupffer cells triggered the phosphorylation of NF-κB p65 upon exposure to conditioned media from APAP-activated hepatocytes. Furthermore, rhHMGB1 and rhHSP70, as surrogates of DAMPs, also enhanced the transcriptional activity of NF-κB in ImKC cells harboring a luciferase reporter. Treatment with IKKβ inhibitors consistently attenuated the conditioned media-induced phosphorylation of NF-κB p65 in Kupffer cells, as well as rhHMGB1- or rhHSP70-induced transcriptional activity of NF-κB in ImKC cells.

In the current study, small-molecule inhibitors of IKKβ activity resolved the DAMP-induced activation of NF-κB activity via IKKβ in the liver of APAP-challenged mice. IKKβ-deficient (*IKKβ*^−/−^) mice exhibit embryonic lethality with massive liver degeneration [35]. Whether conditional ablation or overexpression of IKKβ in the liver can alter APAP-induced hepatotoxicity remains to be defined. Hepatocyte-specific knockout of the IKKα gene (*IKKα^Δhep^*), encoding another catalytic subunit of the IKK complex, represses APAP-induced hepatic injury and mortality; on a molecular level, the NF-κB-inducing kinase (NIK)–IKKα–ROS–ferroptosis axis is involved in eliciting this response [14]. Genetic inactivation of IKKβ in both hepatocytes and hematopoiesis-derived Kupffer cells decreases the incidence of diethylnitrosamine-induced hepatocellular carcinomas due to the reduced expression of tumor promoting cytokines in Kupffer cells [36]. In contrast, mice with hepatocyte-targeted ablation of IKKβ (*IKKβ^Δhep^*) are hyper-susceptible to hepatocarcinogenesis, in which the NF-κB-dependent induction of pro-survival and anti-oxidant genes is impaired [37]. Mice expressing the constitutively active form of IKKβ (*IKKβ-CA*) in hepatocytes do not aggravate chemical carcinogenesis in the liver even though they have altered liver organogenesis during embryogenesis, such as vasculature with enlarged sinusoids and the increased deposition of red blood cells in the liver [38].

The NF-κB heterodimer, consisting of p50 and p65 subunits, is mainly responsible for the canonical transcriptional activity of NF-κB [13]. Mice with constitutive knockout of NF-κB p65 (*NF-κB p65*^−/−^) die during embryonic development due to massive apoptosis in the liver, but are born with normal livers after additional ablation of TNF receptor 1 (*TNFR-1*^−/−^) [39]. Conditional knockout of NF-κB p65 in hepatocytes (*rela^F/F^AlbCre*) sensitizes mice to TNF-induced lethal liver injury, in which susceptibility to TNF in NF-κB p65- or IKKβ-ablated hepatocytes correlates with the loss of NF-κB activity [40]. It remains to be clarified whether conditional ablation of NF-κB p65 in the liver plays a protective or exacerbating role in APAP-induced hepatotoxicity. NF-κB p50-deficient (*NF-κB p50*^−/−^) mice survive and do not show altered APAP-induced hepatotoxicity, suggesting that p50 may be complemented with another subunit of NF-κB [41]. Mice with inducible inactivation of NF-κB p65 in hepatocytes (*rela^F/F^AlbCre*) do not show enhanced liver damage after hepatectomy, the surgical resection of approximately two-thirds of the liver, but exhibit an accelerated cell cycle progression without affecting liver mass generation [42].

## 5. Conclusions

Taken together, small-molecule inhibitors of IKKβ activity limited sterile inflammation in APAP-induced hepatotoxicity. This mechanism protected the APAP-challenged mice from necrotic injury in the liver, and thus rescued the mice from hepatic damage-associated lethality. Finally, we propose that blockade of IKKβ activity in sterile inflammation can mitigate APAP-induced hepatic injury and failure in mice, and shows a potential to complement NAC treatment to protect against APAP overdose.

## Figures and Tables

**Figure 1 pharmaceutics-15-00710-f001:**
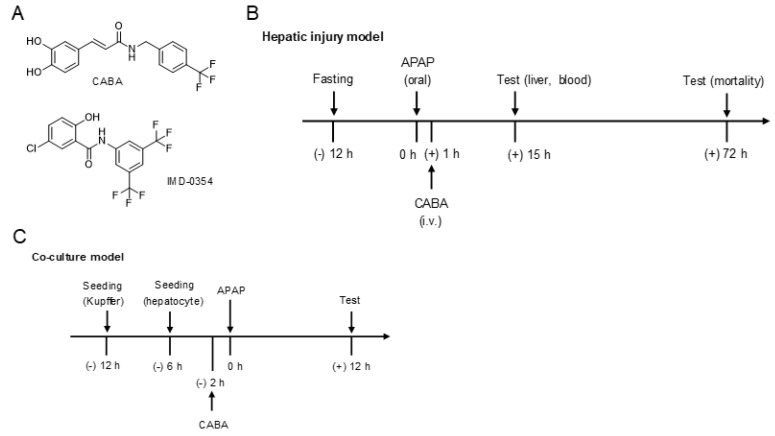
(**A**) Chemical structures of CABA and IMD-0354. (**B**) Experimental protocol of APAP-induced hepatotoxicity in mice. C57BL/6 mice were intravenously treated with CABA at 1 h after oral administration with APAP. (**C**) Experimental protocol in co-culture model. Co-culture between primary hepatocytes and Kupffer cells was pretreated with CABA for 2 h and stimulated with APAP for 12 h in the presence of CABA.

**Figure 2 pharmaceutics-15-00710-f002:**
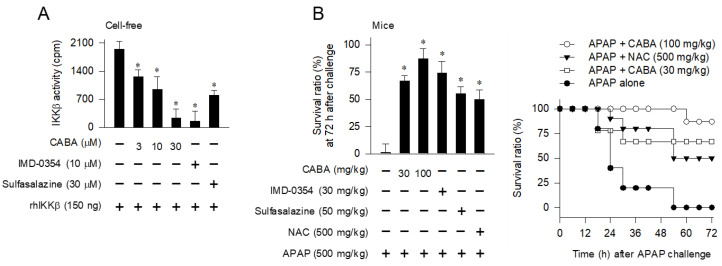
Effects of IKKβ inhibitors on IKKβ activity and APAP-induced mortality. (**A**) Catalytically active rhIKKβ was treated with CABA for 10 min in cell-free reactions. In vitro kinase activity was determined by the incorporation of [^32^P] onto MBP as exogenous substrate from [γ-^32^P]-labeled ATP, and represented as count per min (cpm). * *p* < 0.05 vs. IKKβ alone. (**B**) C57BL/6 mice were treated with CABA at 1 h after APAP overdose. Survival ratio was examined until 72 h. Three independent tests were performed with 9 to 11 mice per group. * *p* < 0.05 vs. APAP alone.

**Figure 3 pharmaceutics-15-00710-f003:**
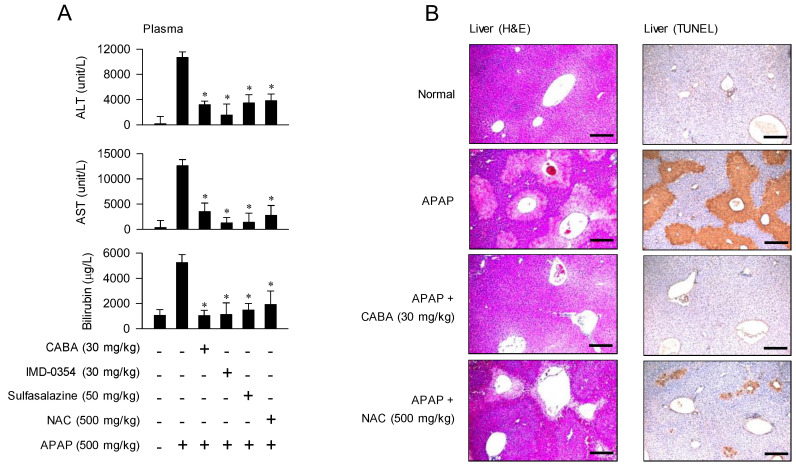
Effect of IKKβ inhibitors on APAP-induced hepatic injury. C57BL/6 mice were treated with CABA at 1 h after APAP overdose. (**A**) Blood samples were prepared at 15 h, and used to measure the levels of ALT, AST, and bilirubin. * *p* < 0.05 vs. APAP alone. (**B**) Liver lobules were biopsied after 15 h. Liver tissues were sectioned into sections with a thickness of 5 µm, stained with H&E or reacted with the TUNEL assay kit, and then examined under microscope. Black scale bars are 200 µm.

**Figure 4 pharmaceutics-15-00710-f004:**
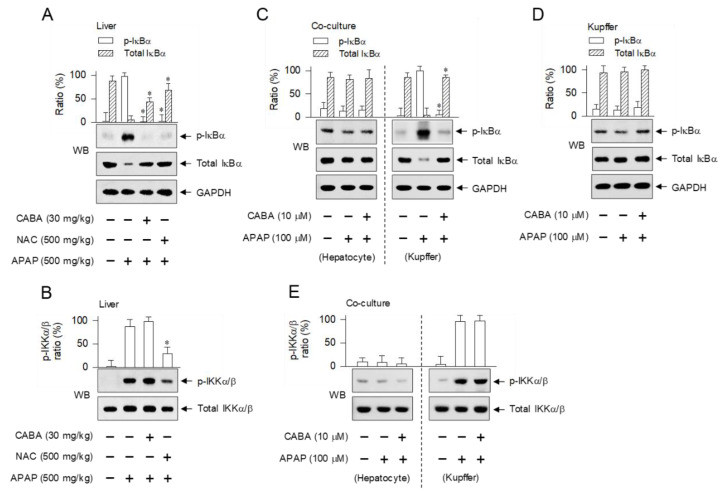
Effect of CABA on APAP-induced activation of IKKβ. (**A**,**B**) C57BL/6 mice were treated with CABA at 1 h after APAP overdose. Liver lobules were biopsied at 15 h. (**C**,**E**) Co-culture between primary hepatocytes and Kupffer cells was pretreated with CABA for 2 h and stimulated with APAP for 12 h in the presence of CABA. (**D**) Primary Kupffer cells were pretreated with CABA for 2 h and stimulated with APAP for 12 h in the presence of CABA. Protein extracts were resolved on an SDS-acrylamide gel by electrophoresis, and subjected to Western blot (WB) analysis with anti-p-IκBα, anti-IκBα, or anti-GAPDH antibody (**A**,**C**,**D**) and anti-p-IKKα/β or anti-IKKα/β antibody (**B**,**E**). Ratio (%) represents relative intensity of signals in WB. * *p* < 0.05 vs. APAP alone.

**Figure 5 pharmaceutics-15-00710-f005:**
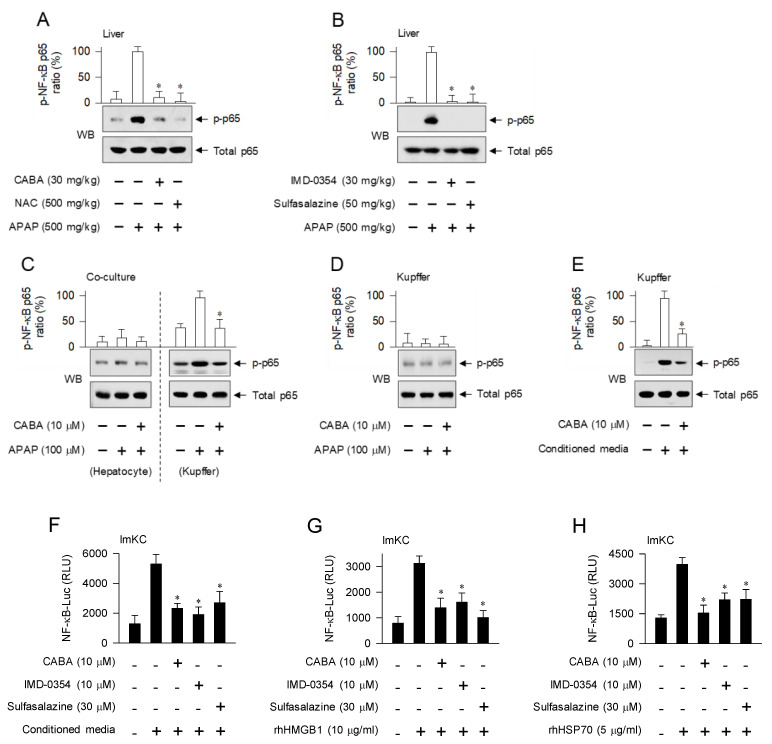
Effect of IKKβ inhibitors on APAP- or DAMP-induced activation of NF-κB. (**A**,**B**) C57BL/6 mice were treated with CABA at 1 h after APAP overdose. Liver lobules were biopsied at 15 h. (**C**) Co-culture of hepatocytes and Kupffer cells or (**D**) monoculture of Kupffer cells was pretreated with CABA for 2 h and stimulated with APAP for 12 h in the presence of CABA. (**E**,**F**) Conditioned media were prepared from hepatocyte cultures after exposure to APAP (100 µM) for 12 h. (**E**) Kupffer cells were pretreated with CABA for 2 h and stimulated with conditioned media for another 12 h in the presence of CABA. Protein extracts were subjected to Western blot (WB) analysis with anti-p-NF-κB p65 or anti-NF-κB p65 antibody. Ratio (%) represents the relative intensity of signals in the WB. (**F**) ImKC cells harboring the NF-κB-Luc reporter were stimulated with conditioned media for 20 h in the presence of CABA. (**G**,**H**) rhHMGB1 or rhHSP70 was excluded from endotoxin lipopolysaccharide by pre-incubation with polymyxin B (20 μg/mL), and used as surrogates for DAMPs. ImKC cells harboring the NF-κB-Luc reporter were stimulated with rhHMGB1 or rhHSP70 for 20 h in the presence of CABA. Firefly luciferase activity, reporting the transcriptional activity of NF-κB, was normalized to *Renilla* activity as a measure of transfection efficiency. * *p* < 0.05 vs. APAP alone, conditioned media alone, rhHMGB1 alone, or rhHSP70 alone.

**Figure 6 pharmaceutics-15-00710-f006:**
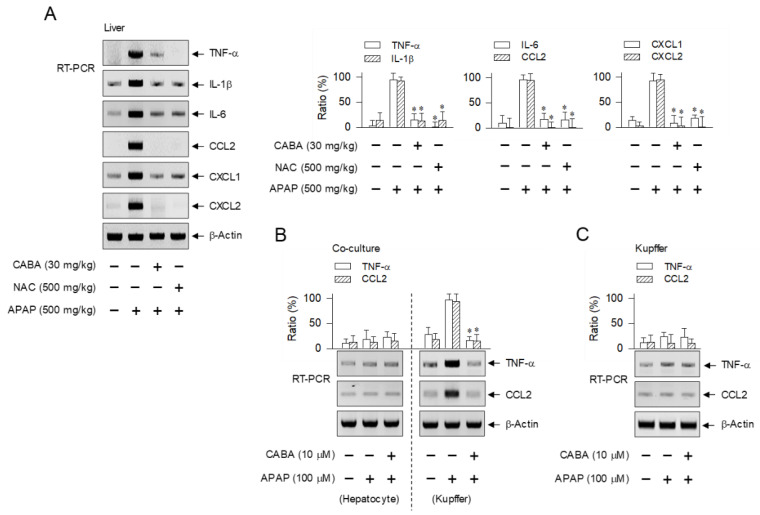
Effect of CABA on the expression of cytokine or chemokine. (**A**) C57BL/6 mice were treated with CABA at 1 h after APAP overdose. Liver lobules were biopsied at 15 h. Total RNAs were subjected to RT-PCR analysis of TNF-α, IL-1β, IL-6, CCL2, CXCL1, or CXCL2 with internal control β-actin. (**B**) Co-culture of hepatocytes and Kupffer cells or (**C**) monoculture of Kupffer cells was pretreated with CABA for 2 h and stimulated with APAP for 12 h in the presence of CABA. Total RNAs were subjected to RT-PCR analysis of TNF-α or CCL2 with internal control β-actin. Ratio (%) represents relative intensity of signals in RT-PCR analysis. * *p* < 0.05 vs. APAP alone.

**Figure 7 pharmaceutics-15-00710-f007:**
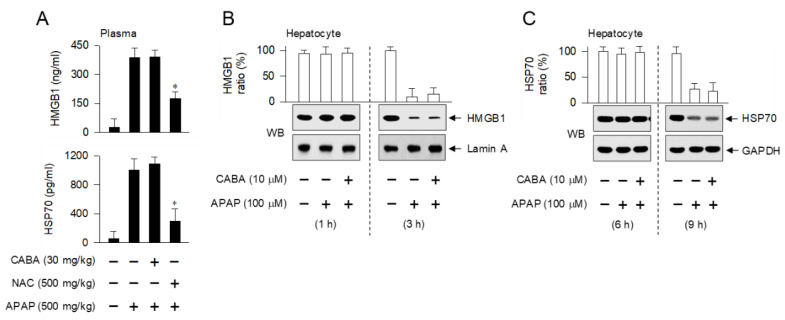
Effect of CABA on the release of DAMPs such as HMGB1 and HSP70. (**A**) C57BL/6 mice were treated with CABA at 1 h after APAP overdose. Blood samples were prepared at 15 h, and used to measure the levels of HMGB1 and HSP70. * *p* < 0.05 vs. APAP alone. (**B**,**C**) Primary hepatocytes were pretreated with CABA for 2 h and stimulated with APAP for the indicated time-points in the presence of CABA. Protein extracts were subjected to Western blot (WB) analysis with anti-HMGB1 or anti-lamin A antibody (**B**) and anti-HSP70 or anti-GAPDH antibody (**C**). Ratio (%) represents the relative intensity of signals in the WB.

**Figure 8 pharmaceutics-15-00710-f008:**
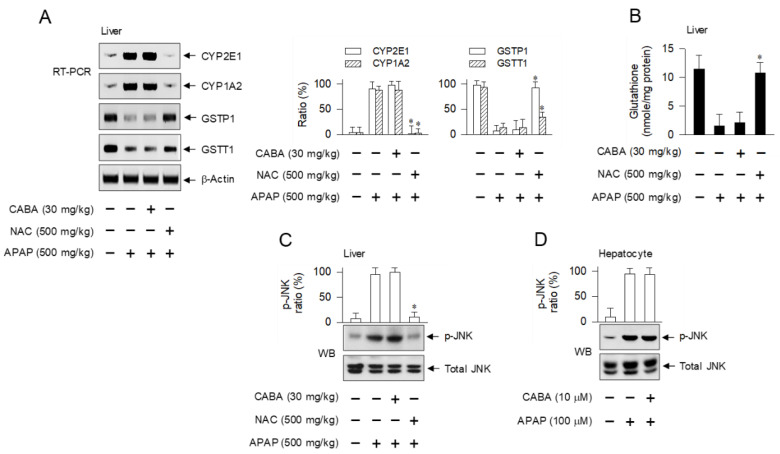
Effect of CABA on APAP-induced oxidative stress in the liver. C57BL/6 mice were treated with CABA at 1 h after APAP overdose. Liver lobules were biopsied at 15 h. (**A**) Total RNAs were subjected to RT-PCR analysis of CYP2E1, CYP1A2, GSTP1, or GSTT1 with internal control β-actin. (**B**) Cell extracts was subjected to ELISA to determine the levels of glutathione, which were normalized to mg of protein. (**C**) Protein extracts were subjected to Western blot (WB) analysis with anti-p-JNK or anti-JNK antibody. (**D**) Primary hepatocytes were pretreated with CABA for 2 h and stimulated with APAP for 12 h in the presence of CABA. Protein extracts were subjected to WB analysis with anti-p-JNK or anti-JNK antibody. Ratio (%) represents the relative intensity of signals in the WB. * *p* < 0.05 vs. APAP alone.

**Figure 9 pharmaceutics-15-00710-f009:**
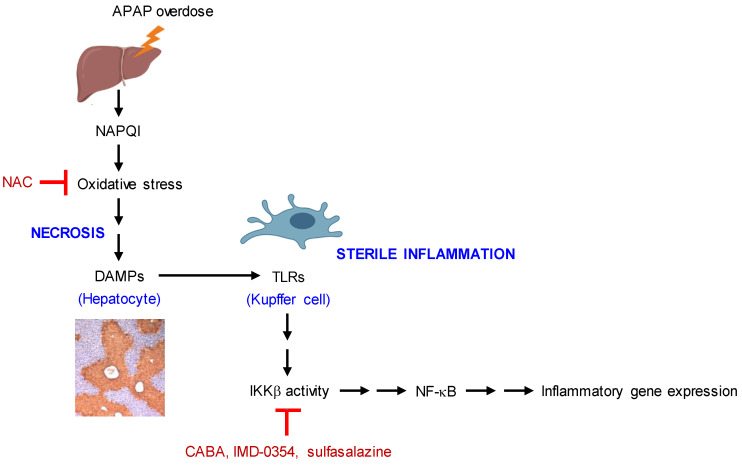
A proposed mechanism of IKKβ inhibition as an antidote against APAP overdose.

## Data Availability

Data will be made available from the corresponding author upon reasonable request.

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
