# Peer review of "Targeting IKKβ Activity to Limit Sterile Inflammation in Acetaminophen-Induced Hepatotoxicity in Mice"

_pharmaceutics, 2023, doi:10.3390/pharmaceutics15020710_

Round 1
Reviewer 1 Report
Acetaminophen (APAP) overdose causes hepatotoxicity. To understand the importance of IKKbeta activity in sterile inflammation during hepatotoxicity, C57BL/6 mice were treated with IKKbeta-inhibitors, caffeic acid benzylamide (CABA), IMD-0354, or sulfasal-azine, after APAP overdose. The authors found that IKKβ inhibitors directly interrupted sterile inflammation in Kupffer cells, although they did not affect the upstream pathogenic events as N-Acetyl cysteine (NAC) dose. The blockade of IKKbeta activity in sterile inflammation can mitigate APAP-induced hepatic injury and may have the potential as a supplement with NAC. The manuscript is well-written and the results are interesting.
Some minor points
1. In Figures 2, 3, and 5, their titles should be ‘Effect of IKKbeta inhibitors on ….’, since they include the results with IMD-0354 and sulfasalazine as well.
2. In Figures 4-8, it is not quite clear what the ‘Ratio (%)’ of the vertical axis means. To what are the ratios? Please describe it in the legend of each figure.
3. Delete a sentence in lines 16-17 on page 6. It belongs to Materials and Methods.
Author Response
Reviewer #1
Acetaminophen (APAP) overdose causes hepatotoxicity. To understand the importance of IKKbeta activity in sterile inflammation during hepatotoxicity, C57BL/6 mice were treated with IKKbeta-inhibitors, caffeic acid benzylamide (CABA), IMD-0354, or sulfasal-azine, after APAP overdose. The authors found that IKKβ inhibitors directly interrupted sterile inflammation in Kupffer cells, although they did not affect the upstream pathogenic events as N-Acetyl cysteine (NAC) dose. The blockade of IKKbeta activity in sterile inflammation can mitigate APAP-induced hepatic injury and may have the potential as a supplement with NAC. The manuscript is well-written and the results are interesting.
Some minor points
1. In Figures 2, 3, and 5, their titles should be ‘Effect of IKKbeta inhibitors on ….’, since they include the results with IMD-0354 and sulfasalazine as well.
- “Effect of CABA on” is corrected to “Effect of IKKβ inhibitors on”.
2. In Figures 4-8, it is not quite clear what the ‘Ratio (%)’ of the vertical axis means. To what are the ratios? Please describe it in the legend of each figure.
- “Ratio (%)” represents relative intensity of signals in WB or RT-PCR analysis.
3. Delete a sentence in lines 16-17 on page 6. It belongs to Materials and Methods.
- “The lines 16-17 on page 6” are deleted.

Reviewer 2 Report
In this study, authors focused on the effect of IKKβ inhibitors on APAP-induced hepatotoxicity. To understand the importance of IKKβ activity in sterile inflammation during hepatotoxicity, C57BL/6 mice were treated with CABA, IMD-0354 or sulfasalazine after APAP overdose. IKKβ inhibitors exhibited a significant protection from APAP-induced sterile inflammation in C57BL/6 mice and rescued the mice from hepatic damage-associated lethality. Authors also studied the effects of IKKβ inhibitors from a molecular point of view in hepatic cells.
Overall, it is a well-designed study with relevant and convincing results. The state of the art and research objectives were clearly presented in the introduction. The experiment was correctly designed and the data treatment was appropriate. In my opinion, the manuscript can be suitable for publication after further minor changes.
Materials and methods section:
Co-culture model: authors should add more details about the establishment of this model. Authors should provide details particularly about the proportion of cells used and the number of hours of culture to establish the model.
Results section:
Figure 2. How authors determined the CABA doses used in mice? Authors should add few words to explain how did they choose 30 and 100 mg/kg?
Figure 2. Survival ratio should be "numbered" as panel C.
Finally, did authors carry out toxicity studies using IKKβ inhibitors on mice?
Author Response
In this study, authors focused on the effect of IKKβ inhibitors on APAP-induced hepatotoxicity. To understand the importance of IKKβ activity in sterile inflammation during hepatotoxicity, C57BL/6 mice were treated with CABA, IMD-0354 or sulfasalazine after APAP overdose. IKKβ inhibitors exhibited a significant protection from APAP-induced sterile inflammation in C57BL/6 mice and rescued the mice from hepatic damage-associated lethality. Authors also studied the effects of IKKβ inhibitors from a molecular point of view in hepatic cells.
Overall, it is a well-designed study with relevant and convincing results. The state of the art and research objectives were clearly presented in the introduction. The experiment was correctly designed and the data treatment was appropriate. In my opinion, the manuscript can be suitable for publication after further minor changes.
Materials and methods section:
Co-culture model: authors should add more details about the establishment of this model. Authors should provide details particularly about the proportion of cells used and the number of hours of culture to establish the model.
- The proportion of cells used and the number of hours of culture to establish the model are added in “the section 2.5. of Materials and Methods.
Results section:
Figure 2. How authors determined the CABA doses used in mice? Authors should add few words to explain how did they choose 30 and 100 mg/kg?
- As added in section 2.4. of Materials and Methods, optimal doses of IKKβ inhibitors were determined through preliminary experiments.
Figure 2. Survival ratio should be "numbered" as panel C.
- Survival ratio is represented as percentage (%). No panel C is found in Figure 2.
Finally, did authors carry out toxicity studies using IKKβ inhibitors on mice?
- No.
- Effect of CABA on the viability of hepatocytes or Kupffer cells is presented in Figure S3.
